# Prototype of an IoT-Based Low-Cost Sensor Network for the Hydrological Monitoring of Landslide-Prone Areas

**DOI:** 10.3390/s23042299

**Published:** 2023-02-18

**Authors:** Pasquale Marino, Daniel Camilo Roman Quintero, Giovanni Francesco Santonastaso, Roberto Greco

**Affiliations:** Dipartimento di Ingegneria, Università degli Studi della Campania “Luigi Vanvitelli”, Via Roma 9, 81031 Aversa, Italy

**Keywords:** landslide early warning systems, capacitive soil moisture sensors, Internet of Things, ThingSpeak^TM^

## Abstract

Steep slopes covered by loose unsaturated pyroclastic deposits widely dispersed in Campania, Southern Italy, are often subjected to shallow landslides that turn into fast debris flows causing a large amount of damage and many casualties, triggered by heavy and persistent precipitation. The slope of Cervinara, located around 40 km northeast of Naples, was involved in a destructive flowslide between 15 and 16 December 1999, triggered by a rain event of 325 mm in 48 h. Hydrometeorological monitoring activities have been carried out near the landslide scarp of 1999 since 2017 to assess the water balance and to identify major hydrological processes involving the cover and the shallow groundwater system developing in the upper part of the underlying limestone fractured bedrock. Since 1 December 2022, a remotely accessible low-cost network has been installed to expand the field hydrological monitoring. The use of a network of low-cost capacitive sensors, communicating within the domain of Internet of Things (IoT) technology, aiming at dispersed monitoring of soil moisture, has been tested. Specifically, the tested prototype network allows measurements of the soil water content at two different points, communicating through a Wi-Fi-based IoT system using ESP32 boards. The ThingSpeak^TM^ IoT platform has been used for remote field data visualization. Based on the obtained results, the prototype of this IoT-based low-cost network shows the potential to expand the amount of hydrological data, suitable for setting up early warning systems in landslide-prone areas.

## 1. Introduction

Rainfall-induced shallow landslides are worldwide threats, especially in urban areas, causing many casualties, heavy damage to roads and buildings, and notable economic losses each year [1]. Many of these events are destructive, catastrophic, and dangerous, owing to their frequent fast propagation [2,3]. Although short periods of intense rainfall are commonly considered as triggering factors of landslides, a reliable landslide early warning system (LEWS) should also consider the major hydrological processes occurring in the slope, which may predispose it to failure [4]. These processes are nonlinear, occur over a large-scale through the boundaries of the slope, evolve over long timescales, and affect the initial conditions of the slope at the onset of precipitation (predisposing causes) [5].

Much research has been conducted in the past decade in landslides studies, providing new perspectives to better understand and identify the predisposing hydrological conditions that determine the response of slopes to precipitation [4], as well as to develop physically based models capable of reliably predicting landslide occurrence [6,7,8,9,10,11,12].

The triggering of rainfall-induced shallow landslides is favored by the storage of rainwater within the soil cover after intense and persistent precipitation [5]. The storage of water also requires that the drainage mechanisms, spontaneously developing in the slopes in response to the precipitation, are not capable of draining out much of the infiltrating water [8]. Thus, to better investigate how the complex predisposing processes control the mechanism leading to slope failure in terms of water storage, field hydrological monitoring is often still needed.

Many hydrological monitoring campaigns in landslide-prone areas around the world are carried out by field sensors [13,14,15,16,17] or satellite measurements [18,19]. Moreover, some recent applications show that root zone soil moisture is often the most valuable hydrologic information for shallow landslide prediction [20]; so, its dispersed monitoring should be considered by low-cost networks with easy installation and maintenance.

Lately, recent advances in open-source software and hardware technologies show the potential for the development of low-cost programmed electronic microcontroller boards for field research with analog or digital sensors [21,22,23]. The Internet of Things (IoT) deals with the physical world through a network of objects that are connected to the internet and process the collected information automatically [24]. Moreover, in recent years, the design of low-cost microcontroller boards based on IoT technologies has been spreading worldwide and has become increasingly important for the establishment of monitoring networks in landslide-prone areas [25,26,27,28].

Many slopes of the Southern Apennines in Campania, Italy, covered with pyroclastic loose granular deposits laying upon fractured bedrock, are often involved in destructive debris flows triggered after heavy rainfall events, with about 300 mm depth and a duration of 24–48 h [20,29]. The sudden triggering and fast propagation of these landslides makes the monitoring of the precursors (i.e., rainfall, soil moisture, and suction) the only way to develop effective early warning systems to mitigate the relevant risk [30]. The total thickness of the mobilized deposits, mainly consisting of ashes and pumices (initially in unsaturated conditions), is usually a few meters (1.0 m–1.5 m) in the steepest part of the slopes, and it is larger at the foot. The pyroclastic granular deposits, nearly cohesionless, often rest on slopes with an inclination larger than the effective friction angle of the soil [31]. In such conditions, the equilibrium is guaranteed by the contribution of the soil suction to the soil shear strength, and the slope failure is caused by the reduction in the soil suction, owing to wetting during rainwater infiltration [9]. The unsaturated state of the involved soils makes drainage interventions ineffective to prevent landslides. However, understanding the major hydrological processes that develop within the soil covers and the exchanges of water with the underlying bedrock, which may control the predisposing conditions, is still undergoing development.

This study refers to the slope of Cervinara, located about 40 km east of Naples (Campania, Italy), which was involved in a catastrophic debris flow between 15 and 16 December 1999, triggered by a rainfall event of 325 mm in about 48 h, causing damage and casualties [32]. Since 2002, hydrological monitoring activities have been carried out at the Cervinara slope near the landslide scarp [11,15]. Moreover, in December 2017, a new automatic hydrometeorological station was installed, aiming at the quantification of the various terms of the hydrological balance of the slope [14].

This paper proposes an upgrade of the conventional hydrometeorological network, making use of low-cost sensors, based on a communication system within the domain of the IoT. In each node of the network, low-cost sensors are connected by an IoT-based control board and communicate with other nodes, transmitting data by a short-range wireless communication network (i.e., Wi-Fi), allowing node-to-node distance of some tens of meters. The choice of low-cost components for the development of the monitoring network is deliberate, to show that dispersed environmental monitoring is easily affordable. However, the proposed IoT network architecture can be replicated with the use of more expensive electronic components, according to the available financial resources. The deployment of dispersed sensor networks based on the IoT can be useful for setting up real-time monitoring for early warning systems, providing easily manageable information for nonexpert decision makers, such as the personnel of small municipalities in charge of local civil protection actions. Thus, this work provides practical means for future applications for communities threatened by landslides, with limited economic resources, to implement and manage monitoring networks over large areas.

To show the potential of the proposed IoT network, in December 2022, a prototype network was installed in Cervinara, for dispersed measurements of soil moisture in different points located 20 m apart from each other, moving away from the landslide scarp of 1999.

## 2. Materials and Methods

The inclusion of hydrometeorological information is a key feature for understanding the major processes controlling the slope response to rainfall events. Basically, new hydrological monitoring activities at the slope of Cervinara have been designed for extending the measurements previously carried out with a new prototype of a low-cost sensor network based on the IoT.

Despite the advancements in communications and internet technologies, most conventional monitoring stations do not collect and publish data automatically on public clouds (such as for IoT applications), even if the community claims live data to enhance the effectiveness of landslide early warning systems (LEWS).

The uploading of live data, which are collected by the new low-cost sensors, as well as by the hydrometeorological station operating since December 2017, has been accomplished, to enable remote hydrological monitoring. Specifically, all the measurements carried out at the slope of Cervinara have been transferred to the ThingSpeak^TM^ internet server for the IoT framework, by a Wi-Fi signal installed near the hydrometeorological station. ThingSpeak^TM^ is an IoT analytics platform (Website: https://www.thingspeak.com/, accessed on 21 December 2022) from MathWorks. It easily allows visualization, analysis, and elaboration of live data sources posted in the cloud by one or more IoT devices that are connected to each other [33].

### 2.1. Study Area

This study focuses on the slopes of Mt. Cornito, located near the town of Cervinara, northeast of Naples, belonging to the Partenio Massif in the Southern Apennines (Campania, Italy).

The slopes of the investigated area present a soil cover composed by loose pyroclastic deposits, usually in unsaturated conditions, resulting from various volcanic eruptions from the Vesuvius and the Phlegrean Fields, occurring during the last 40,000 years [29,34]. The soil cover, a few meters thick, consists of different strata of ashes and pumices, overlaying a bedrock formed by Mesozoic–Cenozoic fractured limestone, usually allocating karst aquifers, with a 200 mm/year average deep groundwater recharge [35].

Hydrological monitoring activities at the slope of Cervinara have been carried out since 2002. The last operating monitoring devices, installed in December 2017, consist of an automatic hydrometeorological station located near the scarp of the 1999 landslide. The equipment includes a complete meteorological station, tensiometers, and TDR probes. The acquisition and the storage of the data at an hourly resolution are ensured by a Campbell Scientific Inc. CR-1000 data logger. The monitoring station is powered by a 12 V battery connected to solar panels through a charge controller [14]. In December 2022, the existing equipment was supplemented with dispersed measurements of soil moisture, by establishing a prototype of a wireless network with two low-cost sensors located 20 m apart from each other. All the operating devices were connected to the internet to ensure the real-time availability of the data, according to the paradigm of the IoT (Figure 1).

### 2.2. IoT Upgrade of the Existing Hydrometeorological Station

The first step in designing a monitoring system based on the IoT was made by updating the data communication from the hydrometeorological station installed in December 2017, by sending live measurements to the ThinkSpeak^TM^ cloud through the internet connection. To accomplish a low-cost internet connection, an Orange Pi microcomputer and an external 4G USB-WIFI portable modem were connected to the existing datalogger (Figure 2).

The connection between the Orange Pi and the datalogger CR-1000 was made by an RS-232 cable with a USB adapter. The Orange Pi was powered by a “switching 12 V” port of the CR1000, through an adaptor to 5 V (the input tension of the microcomputer). The microcomputer is powered on only while sending data (every hour) to avoid the overuse of the 12 V battery supplied by solar panels, which powers the entire monitoring station. In this configuration, the external modem was connected to Orange Pi via a USB port to enable internet access by using the 4G public mobile network with a micro sim. The Orange Pi is also equipped with a 4 Gb SD. It can also provide the temporary storage of data when the public mobile network signal is low.

Moreover, an alarm message notification on Telegram channel to a predefined list of users is also sent whenever the power battery consumption falls below the set threshold of the operating voltage for CR-1000 (~10.9 Volts).

The Orange Pi was programmed to operate automatically from the CR-1000 datalogger to send the hourly data of the rainfall, the voltage of battery, the soil temperature, the soil water contents, and the suction on public channels in ThingSpeak^TM^ IoT.

### 2.3. IoT-Based Low-Cost Network

For each measurement point (Figure 1), ESP32 control boards were used. These boards were provided with an integrated antenna working at 2.4 GHz, suitable for the application of the IoT concept to the network (Figure 3).

The default resolution of a single ESP32 board is 12-bits. The devices are powered via a micro-USB port with lead acid batteries of 12 V 2.3 Ah, regulated to 5 V.

Firstly, for this prototype of a hydrological network consisting of only two nodes (Figure 1), the furthest ESP32 microcontroller board (node 2) was programmed to send the data to node 1 located 20 m apart, using a mesh communication algorithm that broadcasted the data between the nodes with a Wi-Fi protocol.

Node 1 receives the data from node 2 via mesh broadcasting and uploads all information to an IoT dedicated platform (i.e., ThingSpeak^TM^). Specifically, to upload field data coming from both remote nodes and the local node, an ESP-8266EX microcontroller was installed in node 1, connected to the ESP32 main controller, capable of receiving the data via serial communication. The internet connection is provided by a 4G USB-WIFI modem, located near the hydrometeorological station.

Thus, all the data collected by the sensors installed at node 2 and node 1 are sent to ThingSpeak^TM^ cloud every hour. After the connection, both devices are set to a power save mode (deep sleep mode). Consequently, a Real-Time Clock (RTC) (specifically the model DS3231) is used to set an alarm to turn on the ESP32 every time. Additionally, a relay module was added to control the power supply to the sensors, reducing the power consumption during the sleep mode. Given the flexibility of the ESP32 and its low power consumption, these microcontrollers simplify the prototyping of a WI-FI-based IoT application.

#### IoT Network Sensors

The prototype IoT network was equipped with soil water content capacitive sensors connected to ESP32 microcontrollers. However, also other sensors could be connected to the nodes of the IoT network, thus allowing any desired hydrometeorological monitoring activity.

Specifically, the capacitive soil moisture sensors used in this work had a 662 K transistor, acting as a voltage regulator component on board, which regulated the supply voltage at a constant 3.3 Volts, even with a non-constant voltage power source (almost any type of battery).

Moreover, these adopted low-cost capacitive sensors use a TL555C-timer integrated circuit to convert the resonance frequency to an analog signal. The output is a voltage signal proportional to the resonance frequency of the LC circuit of the sensor, strongly dependent on the water content of the soil.

To use these sensors for field soil moisture measurements, specific calibration was required. Owing to the dependence of the resonance frequency on the square root of the dielectric permittivity of the bulk soil, the inverse of the output voltage was linearly fit to approximate the volumetric water content [36,37,38].

The nodes of this prototype were also equipped with DS18B20 sensors for soil temperature. This sensor operates within a tension range from 3.0 V to 5.5 V. The sensor was calibrated directly with a curve provided by the manufacturer in Celsius degrees with an accuracy of ±0.5 °C at 25 °C. It was also equipped with a stainless-steel cap that enabled inserting the sensor directly into the soil.

Each electronic component was put inside a hermetically sealed box. The configuration at each node of the mesh including the controlling boards, sensors, relays, voltage regulators, and batteries is sketched in Figure 4. Moreover, node 1 was powered with two batteries, owing to the larger energy consumption of the two processors using the Wi-Fi connection (ESP32 and ESP8266).

The boxes were fixed to a PVC pole 0.5 m above the ground surface, to ensure WiFi connectivity. For the installation of the network sensors, a trench was excavated, and the probes were gently pushed horizontally into its walls at the depth of 0.5 m below the ground surface (Figure 5).

### 2.4. IoT-Based System Architecture

The entire hydrological monitoring system installed at the slope of Cervinara was designed considering three main communication levels: (1) data collection and exchange, (2) internet access to a database on a web server, and (3) data communication and visualization to the users.

The schematic representation of the entire architecture of the hydrological monitoring system based on the IoT framework is shown in Figure 6.

The first level consisted of two layers: the instruments of the main hydrometeorological station (Layer A) and the nodes of the dispersed network (Layer B). Firstly, the CR-1000 datalogger collected all the data from the meteorological and hydrological measurements, and then it exchanged only the information of the rainfall, soil temperature, battery voltage, water contents, and pressures with the Orange Pi (via cable connection).

Differently, the IoT devices of the network (ESP32 boards) collected the data from the temperature and soil moisture sensors. Then, the information was sent from node 2 to node 1 via Wi-Fi communication.

At the central level, all data were sent from the Orange Pi (layer A) and the low-cost Wi-Fi microchip ESP-8266 (layer B) to ThingSpeak^TM^ IoT cloud to enable the real-time remote monitoring. Communication with the existing 4G telephony network was allowed through the internet access provided by the 4G USB-WIFI modem.

Finally, the third level involved the users in charge. They may be either academic researchers, the authorities in charge of risk management, or even the citizens, to which selected information could be made directly accessible. Authorized users could download the entire field dataset on specific channels in ThingSpeak^TM^ cloud (i.e., https://thingspeak.com/channels/1800121-1800122-1663815 (accessed on 21 December 2022) for the hydrometeorological station and https://thingspeak.com/channels/1900126 (accessed on 21 December 2022) for the sensor network).

In principle, monitoring information is accessible from anywhere, by means of dedicated apps through tablets or smartphones. At this level, landslide prediction models and early warning systems could also be implemented.

## 3. Results and Discussion

### 3.1. Calibration of the Capacitive Soil Moisture Sensors

To show the reliability of the measurements made with low-cost devices, the adopted capacitive sensors were calibrated for the studied soil to define an empirical relationship between the volumetric water content and the inverse of the output voltage ($V_{out}{}^{-1}$). To achieve this, soil samples collected on the Cervinara slope were reconstituted in the laboratory with the soil moist tamping technique [39,40], used for reconstructing samples of granular soils at a desired porosity.

To achieve a chosen porosity, a known quantity of soil was put into a hollow cylindrical vessel (equipped with the capacitive probe at the base) of 14.5 cm diameter up to 3 cm height, tamping it in layers of 1 cm. The total initial weight (G) of the sample was estimated using Equation (1).
(1)$$G=\left(1+w\right)\left(1-n\right)\gamma_{s}V,$$
where w is the gravimetric water content, estimated before the beginning of the test as 40.3% by oven-drying three small soil samples taken from the same material used in the test; n is the porosity to be achieved; $\gamma_{s}$ is the specific weight of the soil; and *V* is the volume of the sample. The initial characteristics of the reconstituted sample are shown in Table 1. 

The test was carried out letting the previously reconstituted sample dry by evaporation at an ambient temperature, while the sensor probe output voltage and sample weight were being recorded. Recordings were conducted every 2 h, as the evaporation rate evolved, and they were stopped when the evaporation rate became very small. The evaporated water was simply estimated as the weight loss between readings.

Specifically, knowing the evolution of the water evaporation in the sample, the volume–weight relationships were used to estimate the volumetric water content $\theta_{i}$ at each time step i. The volume of the water fraction $V_{w}$ is:(2)$$V_{W}=\frac{G_{i}-G_{s}}{\gamma_{w}}=\frac{{\;G}_{i}-\left(G-G_{w}\right)}{\gamma_{w}}=\frac{{\;G}_{i}-G+G_{w}}{\gamma_{w}}$$
where $G_{i}$ is the actual weight of the sample, $G_{s}$ is the weight of the solid volume, G is the initial sample weight, and $G_{w}$ is the initial weight of the water fraction.

As $\theta =V_{w}/V$, where $V_{w}$ could be estimated at each measuring step by subtracting the weight of the solid volume $G_{s}$ from the actual weight, the water content $\theta_{i}$ at each time step was estimated with Equation (3):(3)$$\theta_{i}=\frac{V_{W}}{V}=\frac{G_{i}-G_{s}}{\gamma_{w}V}=\frac{{\;G}_{i}-G+G_{w}}{\gamma_{w}V}$$

The results of the calibration, shown in Figure 7, indicated that the free evaporation allowed reaching a soil volumetric water content as small as 0.18. The investigated soil moisture range, 0.18 <θ< 0.48, was representative of the most common field conditions.

The inverse of the voltage was fitted with the least-squares regression, which resulted in the following relationship with *θ*:(4)$$\theta =\frac{2.65}{V_{out}}-0.76$$

The good alignment of the measurement points along a straight line, indicated by the high value of $R^{2}$ = 0.98, was consistent with the expected behavior of the moist soil according to mixing dielectric models [41]. In fact, the output voltage of the capacitive sensor was proportional to the resonance frequency of the capacitor, in turn related to the inverse of the square root of the relative dielectric permittivity of the moist soil, which usually exhibits a linear relationship with the soil volumetric water content [42]. Both the sensitivity and the accuracy of the estimated soil moisture met the requirements of the measurements for large-scale landslide early warning systems, for which the inclusion of spatially coarse or averaged soil moisture information (i.e., not representative of the local values of the soil moisture at landslide locations) has been shown to significantly improve the ability of the systems to correctly predict landslide occurrences [43,44].

### 3.2. Field Monitoring Data

Field monitoring started on 1 December 2022, and it is still ongoing. A period of nearly two weeks was considered for the visualization of the data from this prototype network (Figure 8).

During this time interval, the monitoring of the sensor network has been continuous, despite the relatively low air temperature occurring in December at the altitude of the investigated site (575 m a.s.l.), that could damage the instruments and the batteries. In addition, the installation in sealed boxes has proven to be reliable for Wi-Fi communication between the nodes located 20 m apart from each other, without discontinuous functioning. The hourly trend of monitoring data allows analyzing the soil moisture dynamics, strictly connected to the observed precipitations.

Figure 8a shows the hourly volumetric water content at a 50 cm depth, where the capacitive sensors were placed, estimated through the calibration relationship (4) from the values of the output voltage and the hourly rainfall recorded by the rain gauge (CP) belonging to the meteorological station of the Civil Protection Agency of Campania, not far from the monitored site (Figure 1). Figure 8b shows the hourly soil temperature recorded at the two nodes.

Two rainfall events were observed during the monitored period. In the first case, 18 mm of rainfall occurred in 41 h, while in the second 46.2 mm fell in 80 h, with a maximum intensity of 10 mm/h.

During the first small event, no significant increase in the volumetric water content was observed by any of the probes. Differently, during the longest and largest event, rainwater infiltration reached the soil layer at 50 cm, where a clear increase in the soil water content was detected by both probes.

Moreover, a slow decrease in the water content occurred between the two events, consistent with the nearly absent upward evapotranspiration fluxes (i.e., the vegetation in late autumn is characterized by leafless chestnut trees and no underbrush). Thus, the water uptake from roots, which can be dense at 50 cm depth, remained low due to the dormant vegetation, and the progressive reduction in the soil moisture was likely due to slow vertical gravitational drainage.

Overall, it is quite clear that the obtained results for both nodes confirmed the same response to the meteorological forcing, with similar trends in terms of the water content and temperature, without a significant difference between the two nodes.

Nevertheless, the pyroclastic deposits of Cervinara are characterized by several soil layers with different textural properties [15,31]. In fact, Figure 8a shows significantly different values of water content at the two nodes, likely due to the variable hydraulic properties of the soil throughout the slope. In fact, local heterogeneities play a big role in the rainwater infiltration process. In fact, the variations in soil texture and aggregate size would give rise to areas with different water content. Moreover, the local stratigraphy can also affect the gravitational drainage towards the deeper layers within the soil cover.

In addition, it is worth noting that the temperature at the investigated depth of 50 cm varied between 8.5 °C and 10 °C during the monitored period (Figure 8b), while the calibration relationship (4) was obtained in the laboratory at an ambient temperature (ranging between 19.5 °C and 20.5 °C). The temperature variations slightly affect the dielectric permittivity of the free water stored in the soil pores and the response of the capacitive soil moisture sensor, but temperature differences within 10 °C cause a few percent variation in the sensor output voltage [45]. Thus, the water content values estimated in the field with the specific calibration relationship (4) can be considered reliable.

## 4. Conclusions

The main idea of this study was to propose a prototype of a low-cost IoT-based monitoring network to expand the amount of available field hydrological data for a landslide-prone area.

The communication between two nodes located 20 m apart through Wi-Fi connection protocol was investigated. However, it is possible to replicate the same network architecture with long-distance communication between one or more IoT devices (i.e., LoRa Network), extending the distance between the nodes up to few kilometers [46].

Since in the mountainous area of Campania, the risk of shallow landslides is widely spread, and it is not affordable to instrument hundreds of slopes with expensive monitoring devices, the future attractive perspective might be to obtain the required hydrometeorological information with dispersed low-cost instruments.

Moreover, the proposed network architecture can be enlarged and integrated with other hydrometeorological monitoring devices, such as rain gauges, piezometers, and sensors for the measurement of electrical conductivity, temperature, and the water level in nearby streams. All data could be received on laptops or smartphones, which can be suitable even for a small municipality, allowing real-time monitoring to manage landslide risk by means of early warning systems issuing local alarms. A more dispersed and complete network could be used for early warning purposes in areas subjected to recurrent shallow landslides, such as the slopes of the Partenio Massif around the city of Cervinara.

Indeed, this example may stimulate the development of practical approaches with low-cost sensors to allow risk management in one or more watersheds of large landslide-prone areas, opening new perspectives based on the benefits of the IoT in terms of environmental sustainability.

## Figures and Tables

**Figure 1 sensors-23-02299-f001:**
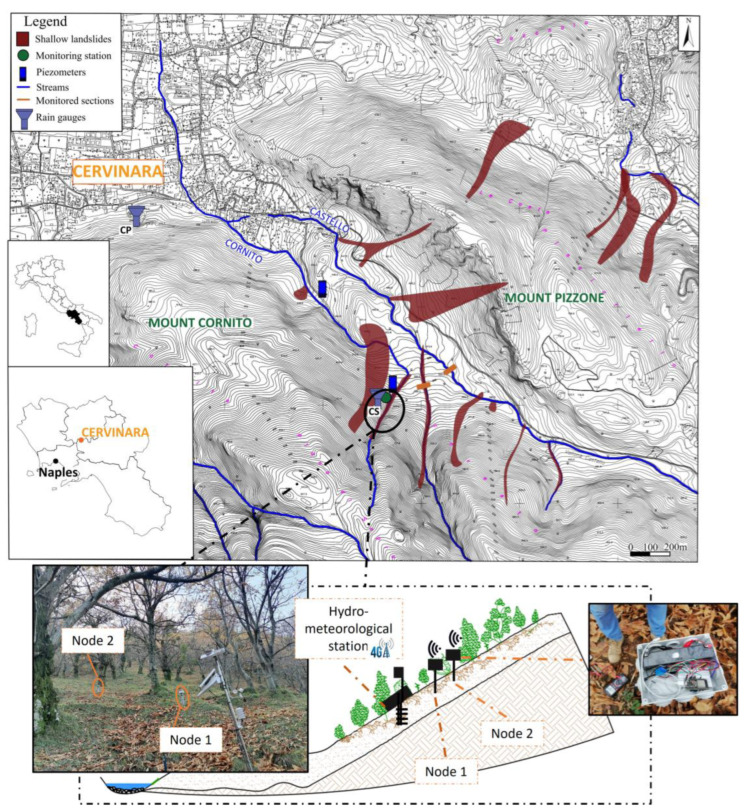
Location of the study area with a closeup view of the position of the installed monitoring IoT devices.

**Figure 2 sensors-23-02299-f002:**
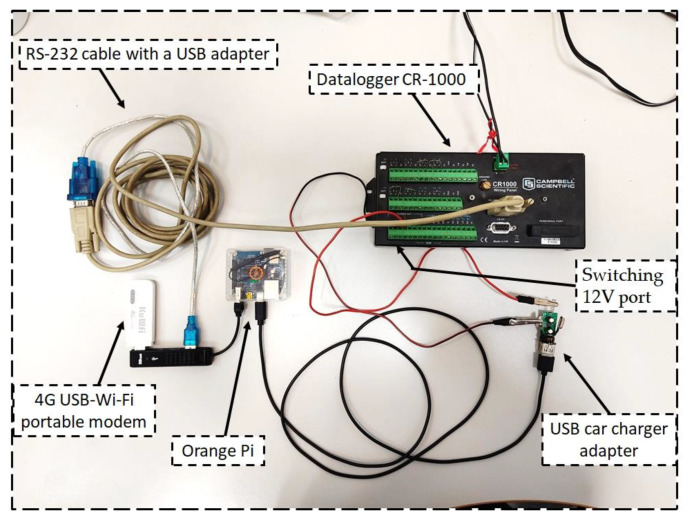
Integration of the internet connection to a conventional datalogger (CR-1000) by adding a control board (Orange Pi) and a portable Wi-Fi router.

**Figure 3 sensors-23-02299-f003:**
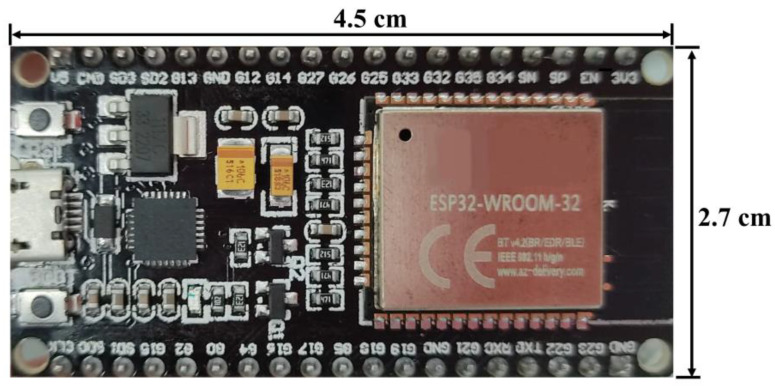
ESP32 Dev kit used to control the system.

**Figure 4 sensors-23-02299-f004:**
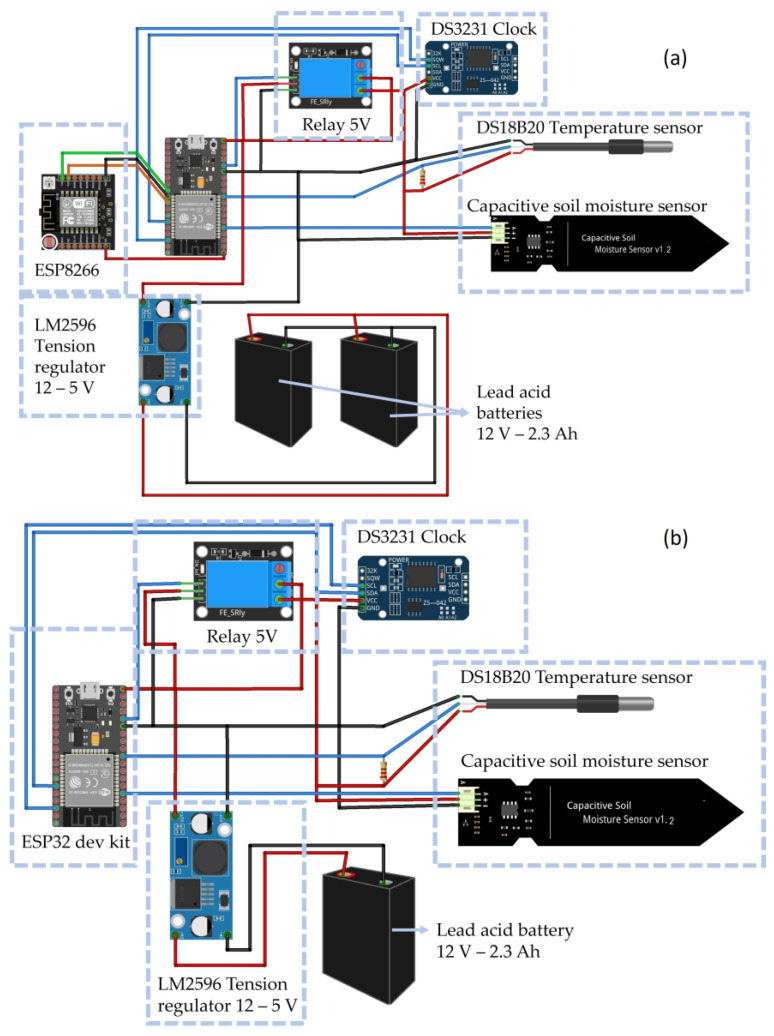
Circuit diagram: node 1 (**a**) with probes, batteries, and microcontrollers ESP32 and ESP8266; node 2 (**b**) with capacitive and temperature probes, battery, and microcontroller ESP32.

**Figure 5 sensors-23-02299-f005:**
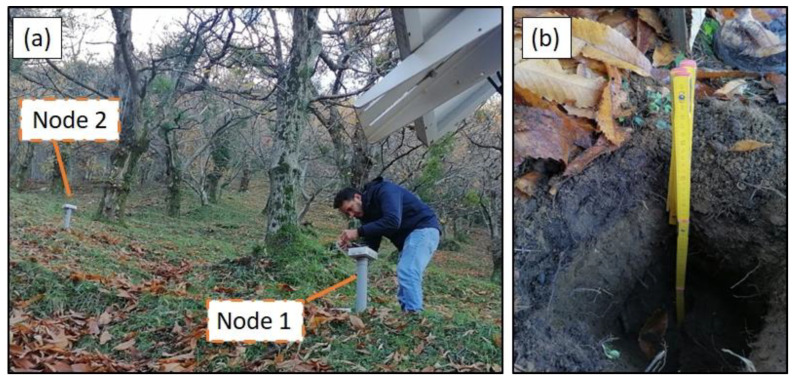
View of the boxes containing the control and communication boards fixed to PVC poles 50 cm above the ground (**a**); excavation where the capacitive probes were horizontally pushed into the soil at 50 cm depth (**b**).

**Figure 6 sensors-23-02299-f006:**
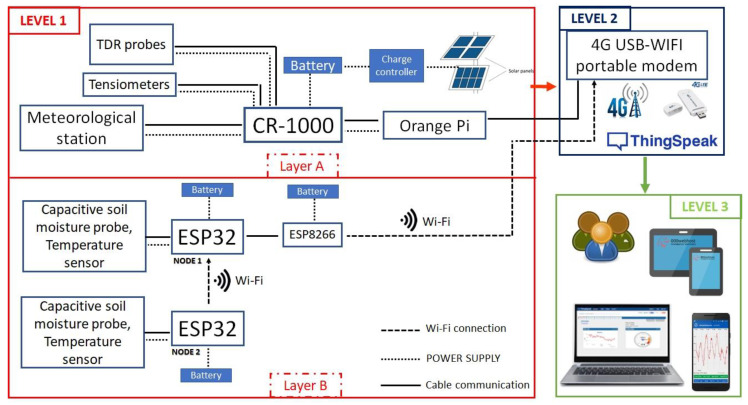
Schematic representation of the IoT framework designed for the monitoring system.

**Figure 7 sensors-23-02299-f007:**
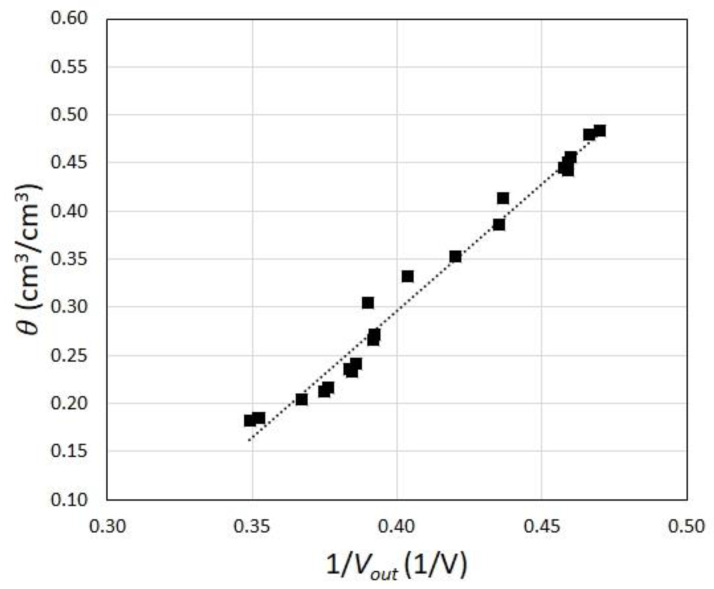
The calibration curve for the soil moisture sensors. The linear fit between the inverse of the voltage *V_out_* and the volumetric water content *θ* (cm^3^/cm^3^).

**Figure 8 sensors-23-02299-f008:**
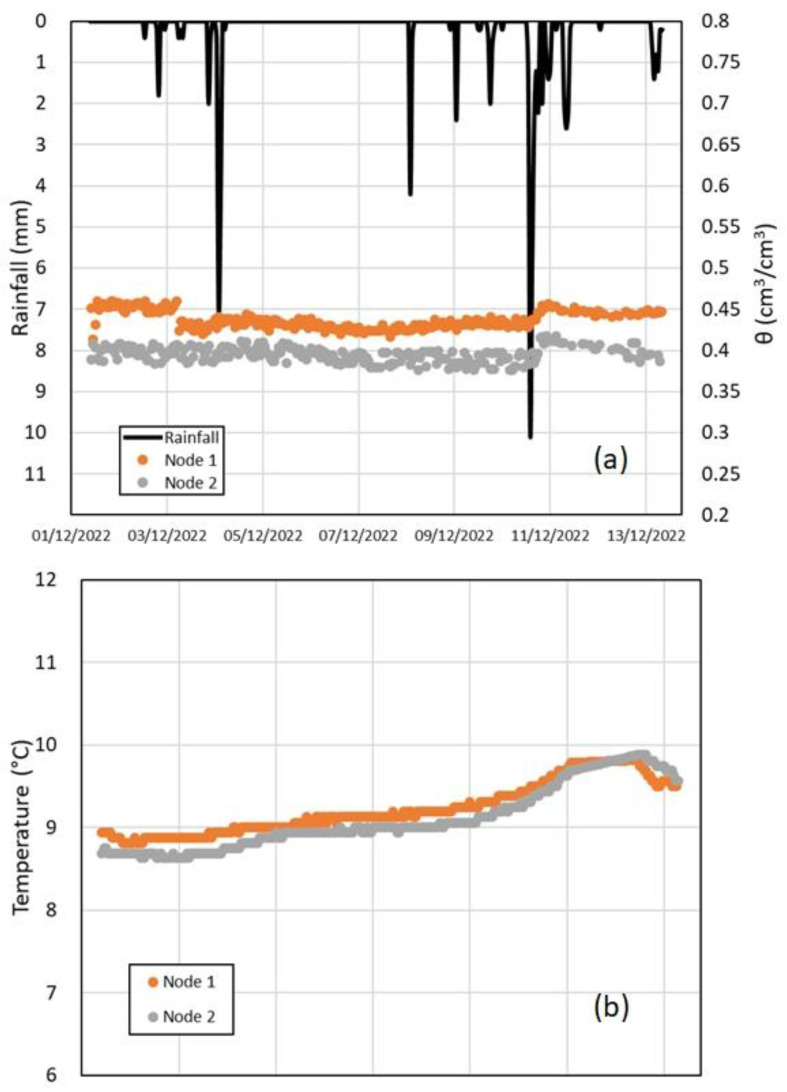
Field monitoring data of the recorded hourly rainfall, volumetric water content (**a**) and temperature (**b**) at a 50 cm depth between 1 and 13 December 2022.

**Table 1 sensors-23-02299-t001:** The general soil properties and the initial conditions of the remolded samples for calibration purposes.

Diameter (cm)	14.6
Height (cm)	3.0
*V* (cm^3^)	502.25
$\gamma_{s}$ (kN/m^3^)	26.2
n (m^3^/m^3^)	0.68
w (g/g)	40.3%
G (g)	602.17
$G_{w}$ (g)	242.84
*θ* initial (m^3^/m^3^)	0.48

## Data Availability

The data collected by the meteorological station (including rainfall, soil and air temperature, relative humidity, solar radiation, atmospheric pressure, and wind speed), water contents, and soil suction and the data of the sensor network may be requested from the authors.

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
