# Peer review of "Prototype of an IoT-Based Low-Cost Sensor Network for the Hydrological Monitoring of Landslide-Prone Areas"

_sensors, 2023, doi:10.3390/s23042299_

Round 1

Reviewer 1 Report

The manuscript presents an interesting design and implementation of a sensor network (based on IoT nodes) for monitoring of landslide-prone areas in Italy.

However, there is no relevant scientific or technological novelty. The proposed prototype is based on widely known IoT implementations, without major contributions from the point of view of sensors and/or electronics. Perhaps the practical application is interesting, but the details are insignificant.

On the other hand, a big mistake made by the authors, is the use of the capacitive soil moisture sensor model SKU:SEN0193. This sensor is totally inadequate for a serious scientific study, even less if the study seeks to correlate soil moisture with another variable of interest, for example rainfall. The used sensor (very low cost) is intended for home and hobby use, but not professional and scientific approach. The sensor is based on an oscillator with the classic LM555, where the soil plays the role of dielectric for the capacitor that is part of the resonator. This sensor is not thermally or electrically compensated. To such an extent that the material used to insulate the electrical circuit will affect the humidity measurement. Also, the resonance frequency is well below the recommended 70Mhz. Above this frequency the conditions of salinity, organic matter, texture and temperature are insignificant and do not affect the measurement of soil moisture. I suggest that the authors do a good review of capacitive type soil moisture sensor technologies and take a good choice of sensors for their studies.

In my opinion, this work must be substantially improved to be published in a high-level journal, such as Sensors.

Reviewer 2 Report

·      

·        Improve the comparison between the measures carrie out with the two different sensors, e.g. calculating metrics as Root Mean Square Error or Nash-Sutcliffe statistical Index. Even if the number of measures are limited for a comprehensive evaluation, these indexes xaan give a quantitative evaluation of the reliability of the adepte sensors

·        Explain better the range of water contente in soil adepte in laboratory for the reconstruction of calibration curve voltage-water content

·        Add more details on possbile uses of the installed sensors: e.g. alarm thresholds, real-time monitoring?

Reviewer 3 Report

Please, read the attached file.

Round 2

Reviewer 1 Report

Dear authors

The work that you carry out is very interesting and commendable, especially for showing the new IoT technologies that allow remote monitoring of variables that support decision making.

In my evaluation, I do not seek to discourage your work and efforts. But the manuscript does not present a significant scientific or technological contribution, which is why it is strange that they try to publish it in a journal as prestigious as "Sensor". These are already known and widely spread technologies. The manuscript may be interesting at a congress in the IoT area.

I consider that the work should be rejected.

In relation to the soil moisture sensor used, I insist that it is for hobbyist use. I am a user and specialist in the use of soil moisture sensors, this sensor can only be used in a pot, to irrigate a garden, it is a low-cost humidity detector, with a short useful life of between 8 to 14 months; totally unfeasible to be used in a long-term monitoring system, it will have to change and calibrate sensors permanently, which makes its use on a large scale and in places of difficult access unfeasible.

Regarding the authors' comment “These risks are widespread in large areas of the world, and the use of cheap components seems attractive to allow the deployment of diffuse monitoring networks.” It is totally wrong. In a data network, the measurement is sought to be as robust and accurate as possible, at a reasonable cost. IoT technologies will help to massify data networks to the extent that they operate correctly, not based on low cost.

I insist, I do not seek to discourage work, but there is no relevant aspect that justifies its publication in a journal.

Reviewer 3 Report

Many thanks for considering all my comments and suggestions.

Author Response

We thank Reviewer#3 for acknowledging the improvement of the manuscript as revised after her/his comments.

Round 3

Reviewer 1 Report

Dear Editor and Authors I repeat my comment and decision to reject the paper. The area of interest and the development proposal is interesting. But, in my opinion, the manuscript does not qualify to be published in a journal of such high prestige as Sensor. It is not a scientific work, nor is there an innovation in IoT. The work is a simple implementation that can be presented in a congress related to IoT, but no more.